# An Optimized Cr(VI)-Removal System Using Sn-based Reducing Adsorbents

George Papadopoulos [1], Theopoula Asimakidou [1,*], Dimitrios Karfaridis [1], Ioannis Kellartzis [2], George Vourlias [1], Manassis Mitrakas [2] and Konstantinos Simeonidis [1]

[1] Department of Physics, Aristotle University of Thessaloniki, 54124 Thessaloniki, Greece; georgpap@physics.auth.gr (G.P.); dkarfari@physics.auth.gr (D.K.); gvourlia@auth.gr (G.V.); ksime@physics.auth.gr (K.S.)

[2] Department of Chemical Engineering, Aristotle University of Thessaloniki, 54124 Thessaloniki, Greece; kellar.john@gmail.com (I.K.); manasis@eng.auth.gr (M.M.)

* Correspondence: tasimaki@physics.auth.gr; Tel.: +30-231-099-8032

**Abstract:** Despite significant risks to human health due to elevated Cr(VI) concentrations in drinking water, a selective adsorbent capable of purifying water before consumption is still not commercially available. This work introduces an integrated household water filtration setup, for point-of-use applications, loaded with a tin-based Cr(VI)-oriented adsorbent that was tested under various contact times, pH values and Cr(VI) concentrations. The adsorbent comprises a chloride-substituted stannous oxy-hydroxide with a structure resembling that of the mineral abhurite. It demonstrated high reducing capacity that triggered the formation of insoluble Cr(III) hydroxides and the complete removal of Cr(VI) in considerably high volumes of polluted water. Test operation of the filtration system verified its ability to produce Cr(VI)-free water in compliance with the impending drinking water regulation, even for extreme initial concentrations (1000 μg/L). Apart from its high efficiency, the potential of the studied material is enhanced by its minimal-cost synthesis method carried out in a continuous-flow reactor by tin chloride precipitation under acidic conditions.

**Keywords:** hexavalent chromium; drinking water; adsorbent; tin oxy-hydroxide; abhurite; filtration

## 1. Introduction

The issue of Cr(VI) intake through drinking water consumption has become the subject of a worldwide public health in the last two decades [1–3]. As a consequence, authorities have been pressured to establish more strict regulations for the verified toxicity of aqueous Cr(VI) species and the frequency of its appearance in water resources [4,5]. This pressure has been seen in actions like those of the U.S. State of California, which was the first to announce a specific regulation limit of 10 μg/L for the Cr(VI) form [6]. However, such decisions are hampered by the absence of a viable technology capable of selectively capturing Cr(VI) from natural water that could be applied in traditional centralized or household water treatment facilities. For this reason, the immediate implementation of the new maximum contaminant level was postponed by the U.S. Federal government on the basis of limited documentation regarding the availability of existing methods to deliver water with Cr(VI) concentrations below 10 μg/L, and the accompanying high purification costs [6].

This argument is partially supported by the failure of most current Cr(VI) removal techniques to comply with the requirements for sufficient water purification at very low concentrations, near-neutral pH values and independency to interferences at an affordable cost [7]. Indeed, treatment methods such as ion exchange, reverse osmosis, electro-coagulation, phyto-remediation and solvent extraction, all fail to meet at least one of the above criteria [8–12]. Satisfactory removal efficiencies are normally

obtained in large-scale facilities with the chemical reduction of Cr(VI) by ferrous salts followed by precipitation [13,14]. However, this method is not always easy to apply as it requires the construction and maintenance of large-scale infrastructure and also produces large volumes of wastes in the form of Cr-loaded sludge [15]. Therefore, adsorption remains the technique of preference especially if materials with high selectivity can be applied. To this end, most of the literature concerns adsorption using activated carbons and other agricultural wastes for Cr(VI) removal referring only to their high specific surface. Unfortunately, these materials provide only very low adsorption capacities and those succeeded at acidic conditions and extremely high concentrations not applicable for drinking water treatment [16,17]. Recent studies have introduced granular adsorbents with high reducing potentials as a class of materials capable of capturing Cr(VI) by reducing it to the insoluble forms of Cr(III) [18,19]. The efficiency of these absorbents is proportional to their ability to operate as electron donors as long as possible. For instance, column tests using $Fe_3O_4$ indicated an adsorption capacity of around 4 mg/g at pH 7.2, whereas $Sn_6O_4(OH)_4$ produced a corresponding value of around 18 mg/g at pH 7.8 maintaining residual concentrations below 10 μg/L [20,21]. These results, which were validated under similar application conditions, suggest that a low-cost adsorbent, highly-selective to Cr(VI), has been realized in the laboratory but is not presently available commercially.

Further research should focus on adsorbents optimization, their validation in real systems, and the comprehensive study of uptake mechanisms, in order to develop treatment systems that support new, stricter, drinking water legislation. This work investigates an upgrade of previously reported Sn(II) oxy-hydroxide adsorbents, and features the partial substitution of their structures' oxygen and hydroxyl groups by chloride ions as a way to enhance Cr(VI) uptake capacity. Adsorbent synthesis was carried out in a continuous-flow reactor to produce large quantities of the granular material whose efficiency was then tested in a typical household filter system using a 9 $\frac{3}{4}$" cartridge.

## 2. Materials and Methods

### 2.1. Synthesis

The synthesis of chloride-substituted stannous oxy-hydroxides was carried out by the hydrolysis of $SnCl_2$ in acidic conditions and a $Cl^-$ rich environment within a two-stage continuous flow reactor similar to the procedure described elsewhere [21]. To establish an excess of $Cl^-$ in the reacting solution, $SnCl_2$ was initially dissolved in 6 N HCl and distilled water to reach a negative pH value. This solution (20 g/L) was then pumped into the first compartment of the reactor (10 L) with a flow rate of 10 L/h and held there for 1 h under vigorous stirring. A second NaOH solution (10% w/w) was continuously introduced into the reactor to maintain a pH value of 2 ± 0.1. The formed precipitate was transferred into the second compartment where the product aged for an additional 1 h. The product's dispersion was collected from the overflow of the second compartment and was then left to sediment and separate from the supernatant water. The solid was washed several times using distilled water and was then dewatered by centrifugation. Finally, the centrifuged sludge was dried at 150 $^\circ$C for 4 h in nitrogen environment to limit $Sn^{2+}$ oxidation and was then milled and sieved to obtain grains of the size range 100–500 μm.

### 2.2. Characterization

The crystalline structure of the tin-based materials was identified by powder X-ray diffractometry (XRD) using a water-cooled Rigaku Ultima Plus instrument with CuKa radiation, a step size of 0.05° and a step time of 2 s, operating at 40 kV and 30 mA. The diffraction patterns were compared to the Powder Diffraction Files (PDF) database [22]. The morphology of the materials was examined using scanning electron microscopy (SEM). Images were obtained in a Quanta 200 ESEM FEG FEI microscope with a field-emission gun operating at 30 kV equipped with an energy-dispersive X-ray spectroscopy (EDS) analyzer. Thermogravimetric analysis (TG-DTA) of the samples was carried out using a water-cooled Perkin-Elmer STA 6000 instrument in the temperature range 50–900 °C,

at a heating rate of 20 °C/min in nitrogen atmosphere. The percentage of $Sn^{2+}$ was determined by acid digestion of the solid followed by titration using $KMnO_4$ solution. A 50 mg sample was dissolved under heat in 20 mL 7 M $H_2SO_4$ and titrated with 0.05 M $KMnO_4$. The end point of the titration was defined by the presence of a persistent weak pink color, indicating that the $MnO_4^-$ ions were no longer being reduced.

A Zetasizer Nano ZS instrument by Malvern Instruments was used to determine the $\zeta$-potential. The isoelectric point was estimated by $\zeta$-potential measurements under various pH values determining the electrophoretic mobility using laser Doppler velocimetry and applying the Smoluchowski approach. All the measurements were performed by dispersing the fine powder of the sample in distilled water at room temperature using 0.001 M $KNO_3$ as the background electrolyte and $HNO_3$ or KOH to adjust the pH. The surface area was estimated by nitrogen gas adsorption at liquid $N_2$ temperature (77 K) using a micropore surface area analyzer according to the Brunauer–Emmett–Teller (BET) model.

*2.3. Chromium Uptake Evaluation*

The efficiency of the produced adsorbent to remove Cr(VI) was preliminary evaluated by batch adsorption experiments carried out using natural-like challenge water prepared according to the National Sanitation Foundation (NSF) standard [23], with pH 7 and residual concentrations in the range of 0–5 mg/L. To prepare 10 L of NSF water, 2.52 g $NaHCO_3$, 0.1214 g $NaNO_3$, 0.0018 g $NaH_2PO_4 \cdot H_2O$, 0.0221 g NaF, 0.706 mg $NaSiO_3 \cdot 5H_2O$, 1.47 g $CaCl_2 \cdot 2H_2O$ and 1.283 g $MgSO_4 \cdot 7H_2O$ were diluted in distilled water. The Cr(VI) stock solution (1000 mg/L) was prepared by diluting reagent grade $K_2Cr_2O_7$ in distilled water. Working standards were freshly prepared by the proper dilution of stock solution in natural-like water. A quantity of 5–100 mg of the fine sample powder was placed into 300 mL conical flasks and dispersed in 200 mL Cr(VI) solutions of concentrations ranging between 0.1 and 10 mg/L. The pH was controlled throughout the experiment by adding either NaOH or HCl of 0.1 or 0.01 M. The dispersion was shaken for 24 h at 20 °C and the solid was separated using 0.45 μm pore-size membrane filters. Residual Cr(VI) concentrations were determined by the diphenylcarbazide colorimetric method. A volume of the filtrate (50 mL) was combined with some drops of the indicator diphenylcarbazide and 3 N $H_2SO_4$. The intensity of the resulting pink color was measured in a Hitachi U-5100 UV-Visible Spectrophotometer at 540 nm. For each condition, experiments were replicated three times.

To obtain a more representative view of the adsorbent's performance, a 9 $\frac{3}{4}$″ cartridge, typically used in point-of-use household filtering systems, was half filled with around 250 g of the tin-based solid. The remaining space in the cartridge was filled with granular activated carbon to serve as a guard in the case of transferred oxidizing ions from tap water ($OCl^-$) that could lower the reducing capacity of the adsorbent. Preliminary tests for possible Cr(VI) uptake by the activated carbon confirmed the practical absence of any contribution. In particular, feeding water spiked with 1000 μg/L Cr(VI) through a filter cartridge filled with granular activated carbon and measuring the concentration in the outflow showed insignificant variations in the initial value in the range ±5 μg/L. The cartridge was placed in the first station of a commercial two-stage filter setup and a polypropylene cartridge with pore size 0.1 μm was placed in the second station to restrain the disintegrated part of tin oxy-hydroxide solids that was periodically observed during preliminary investigation. The setup (Figure 1) was fed with Cr(VI)-spiked tap water at concentrations varying between 100 and 1000 μg/L. To study the effect of different parameters on filter efficiency, experiments were performed with flow rates in the range 5–20 L/h corresponding to empty bed contact times (EBCT) of between 150–30 s and with pH values 7 and 8. For each condition, the procedure was initiated by passing 5 L of test water through the system and then three water samples were collected from the outflow at volume interval of 1 L and measured for residual Cr(VI) as described above. In addition, the presence of solid or dissolved Sn was determined by graphite furnace atomic absorption spectrophotometry using a Perkin Elmer AAnalyst 800 instrument, following sample dissolution in HCl.

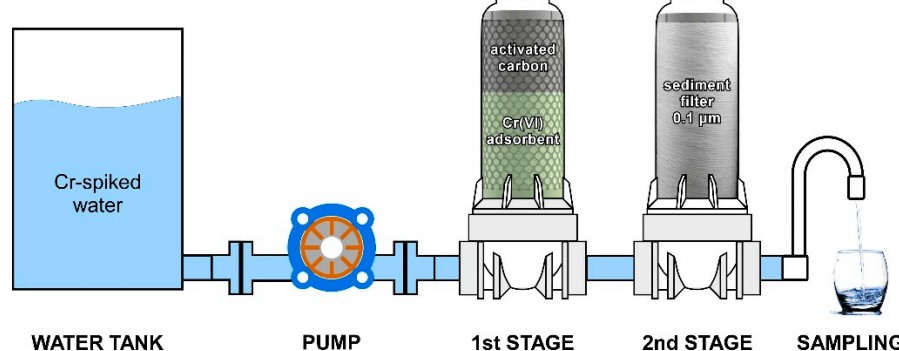

**Figure 1.** Simplified sketch of the point-of-use household filtering system loaded with the developed adsorbent for the removal of Cr(VI) from water.

X-ray photoelectron spectroscopy (XPS) was applied to identify the tin oxidation state in the adsorbent and the speciation of chromium captured on the surface of the adsorbent. The spectrum was acquired in an Axis Ultra DLD system by KRATOS. A monochromated Al–K$_{\alpha 1}$ X-ray beam was used as the excitation source. The pass energy was 160 eV for survey scans and 40 eV for high resolution spectra. The spectra were calibrated in terms of charging induced shifts by assuming the C 1s peak (originating from carbon surface contamination) was located at 284.6 eV.

## 3. Results

### 3.1. Adsorbent Properties

Structural analysis by XRD indicated that the high excess of Cl$^-$ in the reacting mixture enabled the more favorable production of chloro-oxy-hydroxides (Figure 2). In particular, the multiple peaks observed were identified by variations in composition and crystallinity of the Sn$^{2+}$ oxy-hydroxide with the formula Sn$_{21}$Cl$_{16}$(OH)$_{14}$O$_6$, which corresponds to the mineral phase of abhurite (PDF #39-0314). The presence of Sn$_4$Cl$_2$(OH)$_6$ (PDF #15-0675) and an amorphous background were also determined.

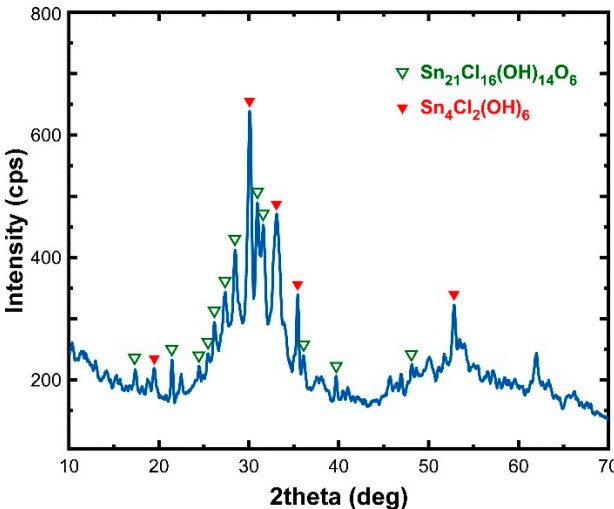

**Figure 2.** XRD diagram of the developed adsorbent.

The morphological characteristics of the precipitated solid resembled those of the bladed structure of abhurite forming a skein-like structure in spherical units with a diameter of around 40 μm (Figure 3a). The spheres normally feature a pit in the one side indicative of their radial growth process. Analysis by energy-dispersive X-ray spectroscopy verifies both the composition homogeneity and the high Cl content of the material. Line scanning element quantification through a sphere suggests an Sn

percentage around 75 wt %, whereas Cl fluctuates around 10 wt %. and O around 15 wt %. (Figure 3b). Such proportions were very similar to those expected for abhurite stoichiometry as well as to the chemical analysis of Sn by atomic absorption spectroscopy and Cl by ion chromatography. In addition, the percentage of tin appearing in the $Sn^{2+}$ oxidation state was measured to be 86% of total tin, which indicates a very high reducing potential of the synthesized material.

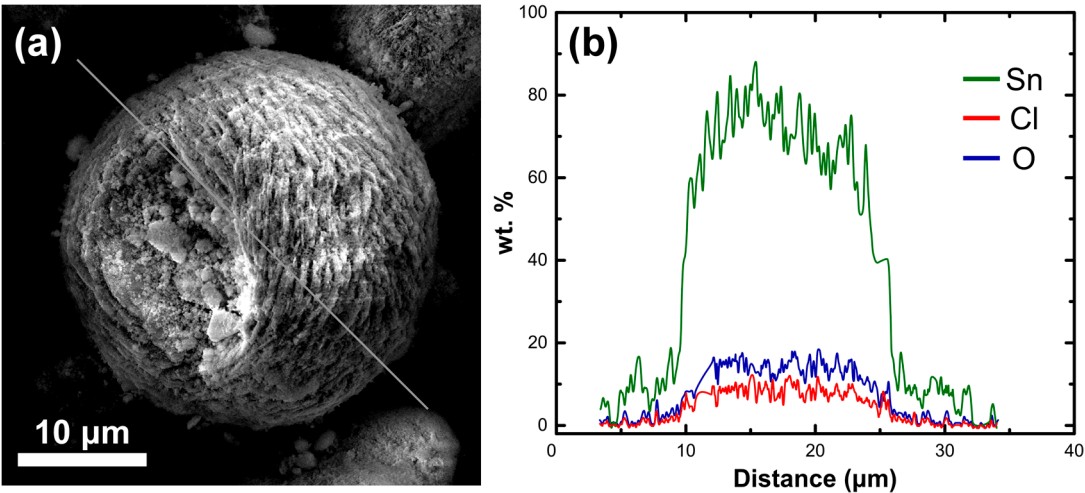

**Figure 3.** SEM image of a primary particle of the adsorbent (**a**) and the corresponding line scan of average element composition (**b**).

The TG-DTA curves shown in Figure 4 also support the chloro-oxy-hydroxide structure of the studied tin-based adsorbent. The weight loss curve was characterized by a small continuous drop (~4%) in the range 100–250 °C, which corresponds to the removal of moisture and physisorbed hydroxyls, and a sharp drop in the range 500–650 °C (~25%), which is attributed to the decomposition of the material after the removal of structural hydroxyls and chlorides in the form of $H_2O$ and $Cl_2$. The last transformation is accompanied by a strong endotherm effect at around 550 °C immediately followed by an exotherm effect at 580 °C signifying the formation of SnO. Due to fluctuations in the average composition and possible variations in crystallinity, the endotherm effect continues up to around 650 °C with smaller intensity.

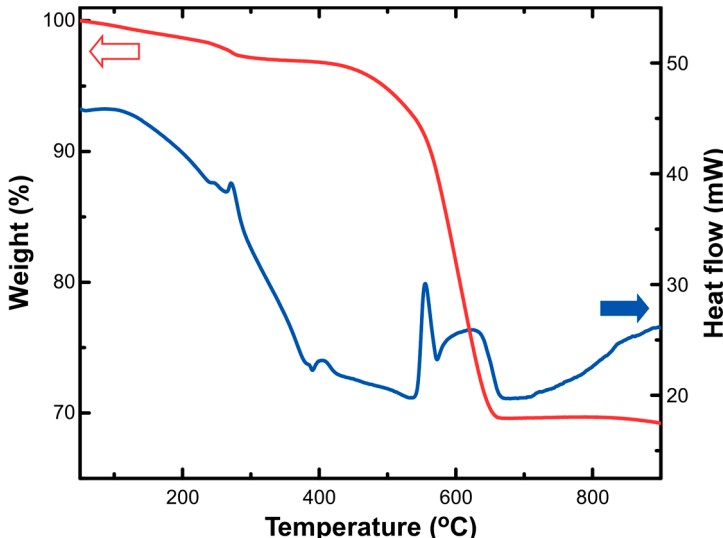

**Figure 4.** Thermogravimetric analysis (TG-DTA) measurements of the developed adsorbent. Arrows indicate the reading axis for the weight loss and the heat flow curves.

Surface properties are a critical feature of the developed material when reviewing adsorbents for water treatment. Here, the specific surface area was found to be 21 m²/g due to the relatively high crystallinity favored by the synthesis conditions. Furthermore, electrophoretic mobility measurements indicate a pH value of 6.1 as the isoelectric point, thus implying a negative surface charging within the normal range of natural water pH (Figure 5).

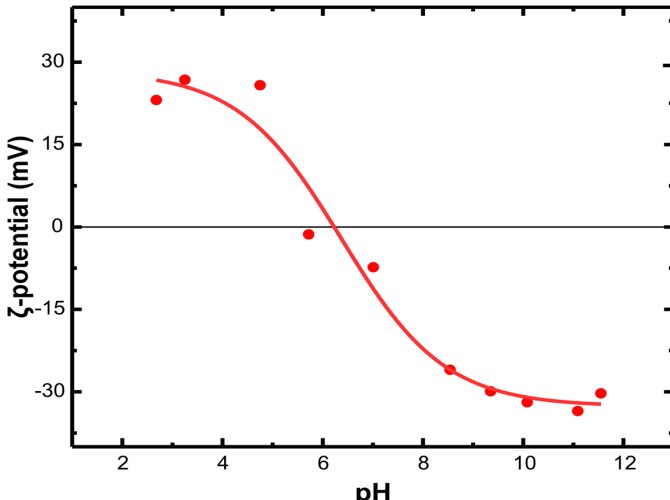

**Figure 5.** Electrophoretic mobility of the developed adsorbent for various pH values.

*3.2. Adsorption Evaluation*

A preliminary estimate of the Cr(VI) uptake efficiency by the adsorbent was obtained by recording the corresponding isotherm after batch adsorption experiments in natural-like water equilibrated at pH 7 (Figure 6). Collected points, spread in a wide residual concentration range of 2–4500 µg/L, were fitted by a Freundlich-type equation $Q = K_F C_e^{1/n}$, with Q denoting the Cr(VI) uptake capacity, $C_e$ the residual concentration, $K_F$ a constant related to the uptake capacity, and n a constant related to the uptake affinity. The calculated values for $K_F$ and 1/n were 14.02 ± 0.83 and 0.096 ± 0.008, respectively ($R^2$ = 0.969). For a residual concentration equal to 10 µg/L (the impending strict regulation limit), the removal capacity meets the value of 17.5 mg/g.

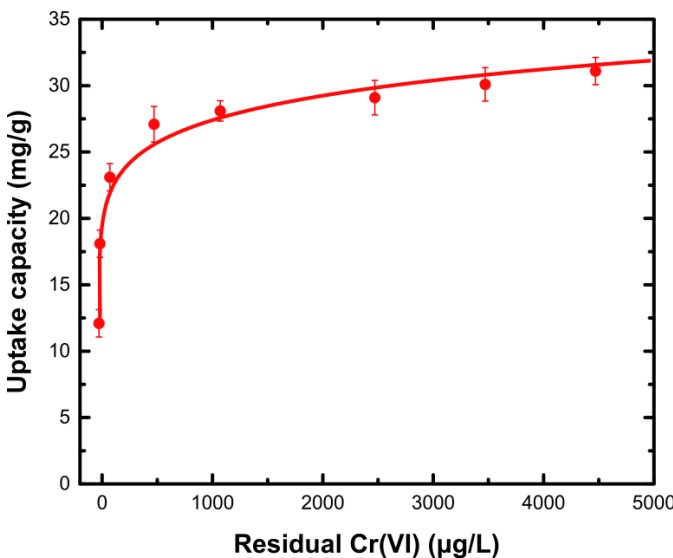

**Figure 6.** Uptake isotherm for Cr(VI) in natural-like water at equilibrium pH 7.

### 3.3. Filter Efficiency

The filter setup for the simulated purification of Cr(VI) polluted tap water was tested under several conditions including various pH values, contact times, and initial Cr(VI) concentrations. On this, the contact time is considered as a more reliable parameter than the flow rate, since it directly reflects the kinetic of adsorption process. For instance, increasing flow rate and keeping the same removal efficiency, as defined by the contact time, is possible by just placing higher quantity of granular abhurite in the cartridge. Specifically, the pH of the feed water was adjusted to the values of 7 and 8 that respectively represent the average natural water acidity and the typical acidity of a Cr(VI)-polluted water that is normally alkaline [24]. For initial Cr(VI) concentrations below 100 µg/L, a value rarely exceeded even in heavily polluted groundwaters, the filter system delivered residual concentrations very close to zero. For this reason, experiments were expanded to include unrealistic high Cr(VI) concentrations of up to 1000 µg/L in order to validate filter efficiency under extreme operation conditions and establish a trend line of its capacity. A summary of the results is given in Figure 7 through the mapping of residual Cr(VI) recorded after numerous experimental runs with various parameters and initial feed volumes. It should be noted that according to the uptake capacity estimated by the batch experiments and the quantity of material used in the filter, saturation of the adsorbent was far to be approached within the scale of its experimental operation and, therefore, no practical modification of its efficiency would be expected.

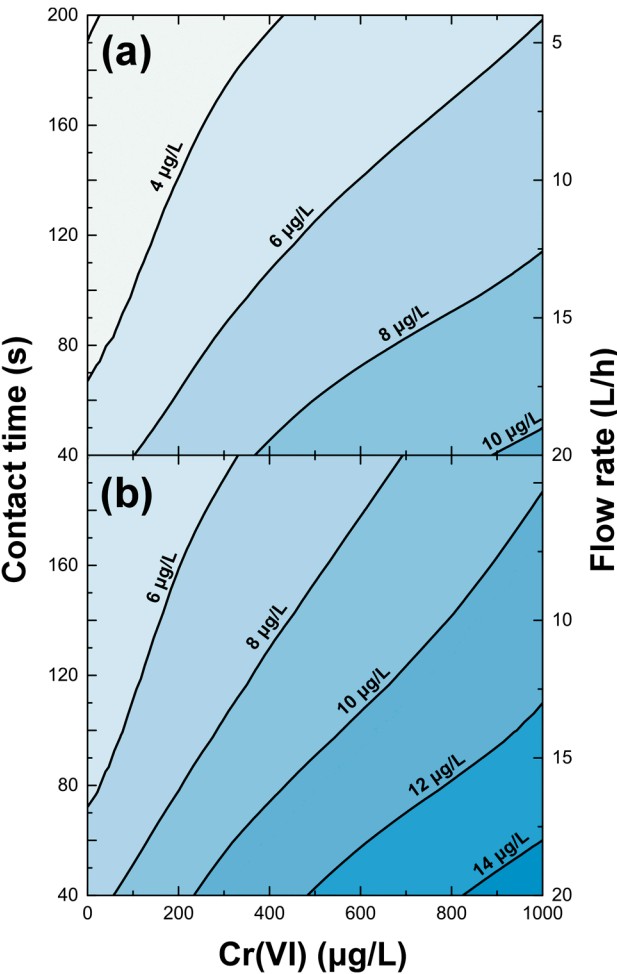

**Figure 7.** Contour plot of residual Cr(VI) versus contact time and initial Cr(VI) concentration in the tested filtering system at pH 7 (**a**) and pH 8 (**b**). Error in Cr(VI) determination was ±2%.

Regarding the upper limit of 10 µg/L, the filter system constantly produced outflows with Cr(VI) concentrations under this value with only minor exceptions observed when testing concentrations above 500 µg/L at pH 8 and with short contact times. Despite the weak effect of operation parameters in the final concentration, indicative of the improved adsorbent's uptake potential and kinetics, it can be concluded that any increase in the initial Cr(VI), the flow rate or the pH have generally a negative role in the efficiency. Importantly, the presence of dissolved Sn in the collected samples was always below 20 µg/L with the 80% of measurements to appear below the detection limit of the analytical method (2 µg/L).

### 3.4. Uptake Mechanism

The reducing potential of the adsorbent as defined by the stabilization of $Sn^{2+}$ in its structure was assumed to be the key for the capture of dissolved Cr(VI). In agreement with titration determination of tin oxidation states, Figure 8a verifies the domination of the bivalent state specifically on the adsorbent's surface even after its use for Cr(VI) removal. Fitting of obtained Sn 3d peaks signified two main contributions at 486.2 eV and 494.6 eV attributed to $Sn^{2+}$ compounds and two smaller contributions at 487.2 and 495.5 eV originating from $Sn^{4+}$ compounds formed on the surface after Cr(VI) reduction or reaction with dissolved oxygen. Quantification of the results suggests that the $Sn^{2+}/Sn^{4+}$ ratio in the few-nanometers depth of XPS measurement was around 4:1.

Analysis of the XPS spectra in the Cr 2p binding range indicates the exclusive presence of Cr(III) compounds in the adsorbent (Figure 8b). Fitting of the $2p_{3/2}$ peak by Gauss-Lorenz functions resulted in the deconvolution of two major contributions centered at 576.5 eV and 577.7 eV, respectively. The first corresponds to a $Cr_2O_3$-type oxide implying the incorporation of Cr(III) into the structure of the adsorbent [25], whereas the second shows the presence of a hydroxide such as $Cr(OH)_3$, which is the product of precipitation, with smaller proximity to the surface [26].

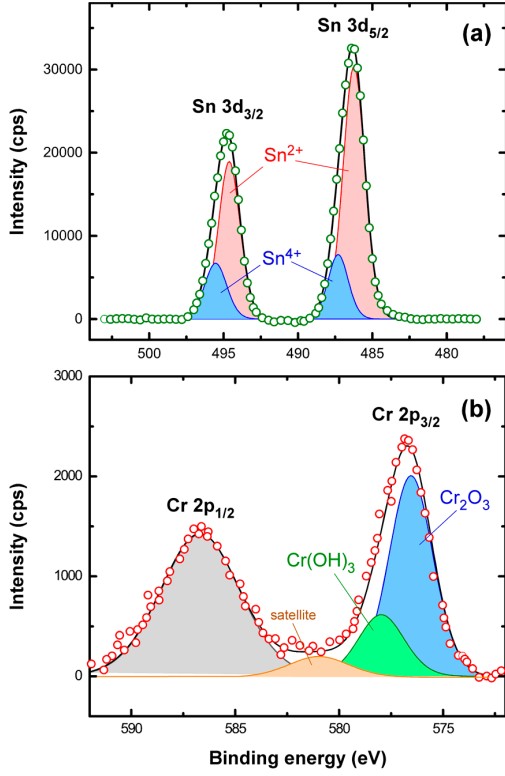

**Figure 8.** Sn 3d (**a**) and Cr 2p (**b**) X-ray photoelectron spectroscopy (XPS) spectra taken for the adsorbent after extended operation for the removal of Cr(VI) from polluted water at pH 7.

## 4. Discussion

The development of an adsorbent comprising synthetic abhurite is demonstrated as a novel approach to selectively capture Cr(VI) in the low concentration range (<100 μg/L) that is frequently met in drinking water resources. Its efficiency is mainly attributed to the high reducing potential of $Sn^{2+}$, representing the dominant tin species (86%), that participates in the electron donation to Cr(VI) and the formation of Cr(III) whether deposited as $Cr(OH)_3$ or even introduced into the crystal structure of the adsorbent, and is in agreement with the conclusions of X-ray absorption fine structure studies on tin oxy-hydroxides [27]. The capacity of the studied adsorbent tested in natural-like water equilibrated at pH 7 and in compliance with a residual Cr(VI) concentration of 10 μg/L, was estimated to be 17.5 mg/g as shown by the adsorption isotherms plotted through batch experiments. Although the obtained value appears to be lower than that previously reported for $Sn_6O_4(OH)_4$ under similar experimental parameters (19 mg/g) [21], one should note that the tin content in the $Sn_{21}Cl_{16}(OH)_{14}O_6$ or $Sn_4Cl_2(OH)_6$ structures identified by XRD is around 73.4 wt %, whereas $Sn_6O_4(OH)_4$ presented a tin stoichiometry of 84.4 wt %. Therefore, considering tin as the cost-defining element of the adsorbent, synthetic abhurite provides an improvement of around 9% in Cr(VI) uptake efficiency. Such performance should be attributed to the significant incorporation of $Cl^-$ into the structure of the adsorbent and the skein-like morphology of its building units both determined by the excess of chloride ions during synthesis under acidic conditions. It seems that apart from the ability of $Sn^{2+}$ to act as an electron donor versus Cr(VI) capture, the presence of $Cl^-$ also enables an ion-exchange mechanism that favors the formation of fresh adsorption sites on the material's surface very similar to the exchange reported between As(V) oxy-ions with sulfate ones during arsenic adsorption from schwertmannite $Fe_{16}O_{16}(SO_4)_3(OH)_{10}\cdot10H_2O$ [28]. At the same time, the peculiar morphology of the material's units built by the combination of layered-structures into spheres facilitates good contact between polluted water and the adsorbent's surface.

Besides the improved efficiency of the Cr(VI) adsorbent, the main breakthrough of this study was the performance of synthetic abhurite within a real filtering system operating in exactly the same way as a typical point-of-use household setup. A relatively short operational period indicated that the filter loaded with the developed adsorbent showed remarkable capabilities to purify water even in the case of extremely high initial Cr(VI) concentrations (1000 μg/L). Particularly, during experiments a total volume of 1 $m^3$ of Cr(VI) spiked water was successfully purified. Considering that only rare occurrences of water with levels of Cr(VI) above 50 μg/L are reported, the testing conditions of this study overestimate the possibility to meet such polluted water but at the same time indicate that the filter will supply clean water with zero Cr(VI) concentration under the typical initial concentrations of 10–50 μg/L. To prolong its operational lifetime, the adsorbent should be protected by an activated carbon guard to avoid direct contact with the oxidizing compounds of water disinfection. According to the estimated uptake capacity of the adsorbent (17.5 mg/g), the expected lifetime of the designed system should be determined after the successful treatment of around 100 $m^3$ of water at pH 7 and initial Cr(VI) concentrations of 50 μg/L, thus implying a lifespan of several years.

## 5. Conclusions

The synthesis of a tin chloro-oxy-hydroxide with a crystalline structure similar to abhurite and improved Cr(VI) uptake efficiency was realized by continuous flow precipitation of $SnCl_2$ under acidic conditions and excess chloride ions. The performance of the developed adsorbent was attributed to the reducing ability of $Sn^{2+}$, which operates as an electron donor for the transformation of Cr(VI) to insoluble Cr(III) species. Due to the substitution of $OH^-$ by $Cl^-$ in structural sites, the material demonstrated a 9% increase in Cr(VI) removal capacity compared to single tin oxy-hydroxides, while its arrangement in skein-like spheres assists easier access of the pollutant to the adsorbent's surface. Testing of the adsorbent in a household filter system verified its ability to eliminate Cr(VI) in the low-concentration range supporting also a good potential for commercial exploitation.

**Author Contributions:** K.S., G.V. and M.M. conceived, designed and supervised the experiments; G.P. performed the synthesis and adsorption experiments; T.A. performed materials characterization measurements and evaluation; D.K. performed XPS measurements and spectra analysis; I.K. carried out BET experiments and data analysis; K.S. and T.A. wrote the paper with the contribution of all co-authors.

**Funding:** This research received no external funding.

**Acknowledgments:** SEM measurements received funding from the EU-H2020 research and innovation program under grant agreement No 654360 having benefitted from the access provided by CSIC-ICMAB in Barcelona within the framework of the NFFA-Europe Transnational Access Activity. Authors would like to thank Anna Esther Carrillo for the experimental assistance during SEM observations.

**Conflicts of Interest:** The authors declare no conflict of interest.

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
