# Peer review of "An Optimized Cr(VI)-Removal System Using Sn-based Reducing Adsorbents"

_water, doi:10.3390/w11122477_

Round 1

Reviewer 1 Report

Synopsis:

The authors investigated the ability of a tin-based adsorbent to remove aqueous Cr(VI) from a simulated household water filtration set-up.  The adsorbent was a chloride-substituted Sn(II) hydr(oxide) with an abhutire-like structure that was synthesized in the lab.  The synthesized material was characterized via XRD, SEM, EDS, TG-DTA, and gas adsorption.  Wet chemical methods were used to determine Sn(II) content.  The Cr(VI) uptake of the adsorbent was evaluated in both batch reactions and in a household system simulation.  In the simulation, the adsorbent (~250 g) was packed into a household water filtering system cartridge with activated carbon.  The loaded cartridge was placed in a set-up with an up-gradient Cr(VI) spiked water source, and a down-gradient sediment filter followed by a sampling outlet.  Adsorbent materials were investigated via XPS after the simulation experiments.  The authors reported an estimated adsorption capacity of 17.5 mg/g based on the adsorption isotherms.  In the household filtration simulations, the authors reported that the adsorbent system was able to reduce Cr(VI) concentrations to 10ug/L or less for spiked waters with Cr(VI) concentrations up to 500 ug/L.  XPS analysis showed that the chromium associated with the adsorbent was exclusively in the form of Cr(III) showing effective chromium reduction by Sn(II).

General Comments:

Overall, I found this work interesting and mostly well-done.  I do have a couple of concerns about the methodology and conclusions at noted below. 

There are a number of grammatical errors that need to be cleared up.

Was any replication done on these experiments?  The text gives the impression that each of the experiments was only run once.

According to Figure 7, the maximum flow rate used in the experiments was 20 L/h.  This is a very low flow rate for a household system.  According to the United States EPA, 1.5 gpm (340 L/h) is a more typical faucets flow rate.  If the goal is to develop a household specific system, was a more typical household flow rate ever investigated?

The authors mention that activated carbon is used in the filter cartridge to guard the tin-based Cr(VI) adsorbent from potential oxidizing agents in the tap water.  However, activated carbon has been studied as an adsorbent for Cr(VI) in water (e.g. Perez-Candela et al. (1995) Water Research, 29 (9); Parlayici et al. (2015) Journal of Nanostructure Chemistry, 5 (3).). Having two potential Cr(VI) loaded into the same canister seems to cloud the results to me.  How much of the Cr removal was due to the activated carbon, rather than the tin hydr(oxide)?  It seems to me that you can’t attribute all of the Cr removal to the tin compound unless you do a thorough post-experiment characterization of the activated carbon to look for sorbed Cr.  Even then, I would think that control experiments with either no activated carbon or with activated carbon but no tin hydr(oxide) would be appropriate.

Longevity of the system?  How does it fare in repeat experiments?

Specific Comments:

Line 108 – What is a “natural-like water”?  Water quality can vary considerably from area to area, and while I understand that some kind of drinking water proxy was appropriate, I think it would be helpful for the authors to indicate the chemistry of a the natural-like water rather than referring the reader to a book, which does not have “natural-like water” listed in the table of contents. 

Line 125-126 – It is stated that a sediment cartridge was placed in the second position to collect any released tin oxy-hydroxides.  Was the cartridge investigated for the presence of released tin-materials?  What about chromium hydr(oxides)? If so, what were the findings?  I did not see this discussed in the text.

Line 225 – Can you quantify “most of the times”?

Line 271 – This is a bit picky on my part, but saying “the developed adsorbent has infinite capabilities to purify water” seems a bit pretentious.

Author Response

Reply to reviewers

Reviewer 1

General Comments:

Overall, I found this work interesting and mostly well-done. I do have a couple of concerns about the methodology and conclusions at noted below. 

There are a number of grammatical errors that need to be cleared up.

The manuscript was revised accordingly by a native speaker of English to improve this part. Changes are indicated with different text color.

Was any replication done on these experiments?  The text gives the impression that each of the experiments was only run once.

Chromium uptake evaluation experiments were repeated three times to obtain the given average. In the revised manuscript this is mentioned in the experimental part and shown by error bars in the adsorption isotherm figure.

Concerning Figure 7 which is a contour plot of three variables deriving from residual concentrations measured for different contact times and initial Cr(VI) concentrations, the graphical representation of error bars is not possible. For this reason, we referred to the accuracy of these measurements within the text. In particular, for the given data, the error was around ±2 % for Cr(VI) determination and less than ±1 % for flowrate.

According to Figure 7, the maximum flow rate used in the experiments was 20 L/h. This is a very low flow rate for a household system. According to the United States EPA, 1.5 gpm (340 L/h) is a more typical faucets flow rate. If the goal is to develop a household specific system, was a more typical household flow rate ever investigated?

Although this comment is meaningful towards application of the described Cr(VI) removal system, we need first to object on the suggested values for typical tap water flowrates as non-realistic for the studied case. In fact, a flowrate of 1.5 gpm is proposed by EPA as the maximum to succeed important saving in the consumption of tap water for bathroom facilities while a flowrate of 0.8 gpm (180 L/h) is also proposed as the minimum to preserve acceptable hygiene. This demand would be valid if we referred to a whole-house filtration system. However, the developed system is introduced as a point-of-use solution oriented to purify just the fraction of tap water used for direct or indirect consumption by human which is only a small percentage of the total water volume needs in the entire house. In addition, its efficiency to capture a very toxic pollutant such as hexavalent chromium and provide completely clean water is considered as the most critical requirement above any other design parameter that can be modified on demand. For point-of-use water filters the general rule is “slow flowrate is always better than faster” to succeed higher purification performance. In fact, restricting water supply rate (and increasing contact time) is one the aims of using fine granules or sub-micron porous tubes for packing material into filter. To this direction, a typical carbon cartridge should operate with a flowrate not higher than around 80 L/h.

Coming back to our study, our intention was to simulate a household filter operation but at the same time test the material under extreme conditions of Cr(VI) pollution and obtain a performance map like the one shown in Figure 7. According to this, the initial Cr(VI) concentration and the contact time were used as the evaluation variables. On this, we consider the contact time as a more reliable parameter than the flowrate since it directly reflects the kinetic of adsorption process. For instance, assuming that 20 L/h is a relative low flowrate, by placing a double height of the granular abhurite in the cartridge operation at the same contact time (40 s) would be possible with a flowrate of 40 L/h. Both cases would bring exactly the same removal efficiency. In other words, one can combine the contact time and the Cr(VI) concentration in order to design the optimum cartridge at any geometry and operation flowrate. We must also clarify that this work does not represent a long-term validation of a commercial product that would only bring a breakthrough curve and the effective lifetime for a typical Cr(VI) concentration e.g. 30 μg/L. That would be also interesting but would require extremely high experiment period (years) and resources not available in laboratory-scale research.

Finally, as mentioned evaluation experiments were carried out for initial Cr(VI) concentration not lower than 100 μg/L. Considering that the majority of water with high levels of Cr(VI) range in concentration below 50 μg/L, with only 2 % of these occurrences to correspond in concentrations 50-100 μg/L, this study overestimates the possibility to meet such polluted water but indicates that under typical conditions (10-50 μg/L) the filter will supply clean water with zero Cr(VI) concentration.

Parts of this discussion was included in the revised manuscript to support the followed methodology.

The authors mention that activated carbon is used in the filter cartridge to guard the tin-based Cr(VI) adsorbent from potential oxidizing agents in the tap water.  However, activated carbon has been studied as an adsorbent for Cr(VI) in water (e.g. Perez-Candela et al. (1995) Water Research, 29 (9); Parlayici et al. (2015) Journal of Nanostructure Chemistry, 5 (3).). Having two potential Cr(VI) loaded into the same canister seems to cloud the results to me.  How much of the Cr removal was due to the activated carbon, rather than the tin hydr(oxide)?  It seems to me that you can’t attribute all of the Cr removal to the tin compound unless you do a thorough post-experiment characterization of the activated carbon to look for sorbed Cr.  Even then, I would think that control experiments with either no activated carbon or with activated carbon but no tin hydr(oxide) would be appropriate.

To be on the safe side we did such preliminary test in advance to exclude the appearance of any contribution from the activated carbon or anything else from the system parts. At such concentrations activated carbon has practically zero effect in Cr(VI) contribution. In particular, flowing water spiked with 1000 μg/L through a pure activated carbon filter cartridge and measuring the concentration in the outflow showed insignificant variations in the initial value in the range ±5 μg/L. For this reason, the whole removal efficiency was exclusively attributed to the tin-based adsorbent.

Concerning existing literature on the efficiency of activated carbons as chromium adsorbent, only a weak chemical affinity can be derived from published results. It is indicative that all works are carried out for initial concentrations higher than 100 mg/L which is at least two orders of magnitude above the working range of our study. Therefore, any comparison between results meaningless. Furthermore, reported efficiencies refer to acidic conditions of testing, which do not fulfill the criteria for evaluation of water treatment processes*, while at higher pH values the obtained adsorption capacity is restricted.

*Criteria for evaluation of water treatment processes

Ability to achieve residual concentration Ce < MCL = 10 μg Cr(VI)/L. Removal capacity at Ce = MCL (Q10 = 17.5 mg Cr(VI)/g). Time of the process lower than 3 min. Evaluation in continuous flow conditions. Ability for full scale implementation. Maintaining the physical and chemical characteristics of water.

Longevity of the system?  How does it fare in repeat experiments?

This is an important question that we already tried to cover in the manuscript but we need to clarify further. During experiments we fed the system totally with around 1 m3 of Cr(VI) spiked water at different concentrations and pH values. When repeating the same condition, the observed residual Cr(VI) showed no significant variation to the previous runs suggesting that no modification of its operation was observed within the testing period. According to the adsorption isotherm, the system is expected to work with the same efficiency up to 100 m3 and this signifies a rather good longevity for a water filter. We should inform the reviewer that at this moment the continuation of operation with Cr(VI) spiked water reached around 4 m3 of successfully treated water without any obvious loss of efficiency. A comment on this was added to the manuscript.

Specific Comments:

Line 108 – What is a “natural-like water”?  Water quality can vary considerably from area to area, and while I understand that some kind of drinking water proxy was appropriate, I think it would be helpful for the authors to indicate the chemistry of a the natural-like water rather than referring the reader to a book, which does not have “natural-like water” listed in the table of contents. 

As explained in the manuscript the called “natural-like water” corresponds to an artificially-prepared challenge water matrix with chemical composition similar to an average groundwater standard as defined by the National Sanitation Foundation (NSF). Therefore, this is a representative medium to study the efficiency of any adsorbent when the major sources of interferences, which would be met in real application, occur. Definitely, evaluation is more accurate than carrying experiments in distilled water.

Indeed the absence of details about the composition of NSF water is confusing to the reader. For this reason we introduced the chemical reagents used to prepare this standard i.e. by diluting 2.52 g NaHCO3, 0.1214 g NaNO3, 0.0018 g NaH2PO4·H2O, 0.0221 g NaF, 0.706 mg NaSiO3·5H2O, 1.47 g CaCl2·2H2O and 1.283 g MgSO4·7H2O diluted in 10 L of distilled water. Then, the corresponding composition of NSF water is HCO3- 183 mg/L, NO3- 2 mg/L, PO43- 0.04 mg/L, F- 1 mg/L, SiO2 20 mg/L, Ca2+ 40 mg/L, Mg2+ 12.7 mg/L, Na+ 88.8 mg/L, Cl- 71 mg/L, SO42- 50 mg/L.

Line 125-126 – It is stated that a sediment cartridge was placed in the second position to collect any released tin oxy-hydroxides.  Was the cartridge investigated for the presence of released tin-materials?  What about chromium hydr(oxides)? If so, what were the findings?  I did not see this discussed in the text.

Yes, during the first experiments collected water from the outflow periodically showed some turbidity attributed to adsorbent’s particles disintegration. By placing a second polypropylene cartridge with pore size 0.1 μm the release particles of adsorbent were collected and kept contributing in Cr(VI) removal. Concerning chromium hydroxides, their precipitation takes place on the adsorbent’s surface and as XPS studies suggest they appear there strongly attached. The release of precipitated chromium from the first stage or even chromium hydroxide re-dissolution were excluded by proper measurements in the outflow of first filter.

Line 225 – Can you quantify “most of the times”?

This general expression implies to 80 % of measurements appearing below the detection limit.

Line 271 – This is a bit picky on my part, but saying “the developed adsorbent has infinite capabilities to purify water” seems a bit pretentious.

Yes, the reviewer is right. We modified the sentence and especially the term infinite.

Reviewer 2 Report

Detailed comments are in the attachment

Author Response

Reply to reviewers

Reviewer 2

Introduction should be supplemented with Cr(VI) adsorption on other sorbents (e.g. activated carbons)

A brief outline of existing literature about other adsorbents tested for the removal of Cr(VI) of was included in the introduction.

Complete the filtration process methodology (how often and after what volume of residual chromium was determined, for which initial concentrations)

The requested experimental details was included in the revised version of the manuscript.

On what basis was it assumed that activated carbon removed only OCl-?

This issue was already analyzed in a similar comment by the first reviewer. Preliminary tests were taken to exclude the appearance of any contribution from the activated carbon or anything else from the system parts. Activated carbon was found with practically zero effect in Cr(VI) contribution.

Complete Fig. 4 (axis values and change the legend)

The axis values for heat flow were placed in the revised figure while the corresponding reading axis for each curve was explained in the caption.

How good was fit isotherm Freundlich

We apologize for omitting these information. Error values for KF and 1/n and the R2 of the fitting are now included in the text.

Fig. 7 is illegible, please put experimental points

We would kindly ask the reviewer to understand that Figure 7 is a graphical representation correlating three variables in a two-dimension plot forming isobar curves and therefore cannot include the experimental points. The alternative way is to present the data by a three-dimensional plot (see figure at the end of paragraph) but then it would be very difficult for the reader to realize the outcome of the experiments. On the opposite, the supplied contour plot, generated by several runs of the setup and sufficient replications, works as a performance map to identify the outflow concentration by selecting the initial Cr(VI) concentration and the contact time.

Has there been clogging of the adsorption bed with Cr (III) ions?

The precipitation of chromium hydroxides takes place on the adsorbent’s surface and as XPS studies suggest they appear there strongly attached. Their estimated quantities are orders of magnitude lower than the adsorbent’s mass and therefore, clogging is not very possible to occur. In practice, after treatment of around 1 m3 of polluted water no loss of flowrate was observed.

Line 270-272 - what does it mean - infinite possibilities.

Indeed this term is inaccurate and replaced.

It is premature to suggest that the system may work in a household for several years

Although the total lifetime of the filter was not validated by long-term use, to our opinion there are sufficient indications to assume this saying. First, our previous studies on a similar tin-based adsorbent for Cr(VI) removal (Kaprara et al., Sci. Total Environ. 2017, 605, 190–198) showed that small-scale column tests with granular material provide better capacity than the one expected from the adsorption isotherms of batch tests. So, a column operation like that of our filter system should at least succeed the efficiency of the corresponding isotherm and treat e.g. around 100 m3 of water with 50 μg/L Cr(VI) which correspond to more than 10 years of effective operation when water is used for consumption and cooking (5 L/day/person). Second, no loss of efficiency was observed during the experiments up to this moment having reached the treatment of around 4 m3 of heavily polluted water.

Conclusion 286 - 288 has no confirmation in the studies – correct conclusions

We believe that an adsorbent able to provide a capacity of 17.5 mg/g in the concentration range below 10 μg/L validated by column tests is very probable to achieve commercial promotion in the near future. Besides, when this material is oriented to the uptake of toxic Cr(VI) for which no specialized adsorbent is available the potential becomes larger. Since the reviewer considers this approach as inordinate we modify the sentence

Round 2

Reviewer 1 Report

The authors have addressed the concerns I had about the original manuscript. 

Reviewer 2 Report

The corrections made by the authors and the completed manuscript are acceptable